# Adenomatous polyposis coli-binding protein end-binding 1 promotes hepatocellular carcinoma growth and metastasis

**Takeshi Aiyama**[1], **Tatsuya Orimo**[1], **Hideki Yokoo**[1]\*, **Takanori Ohata**[1], **Kanako C. Hatanaka**[2], **Yutaka Hatanaka**[2], **Moto Fukai**[1], **Toshiya Kamiyama**[1], **Akinobu Taketomi**[1]

**1** Department of Gastroenterological Surgery I, Hokkaido University Graduate School of Medicine, Sapporo, Hokkaido, Japan, **2** Department of Surgical Pathology, Hokkaido University Hospital, Sapporo, Hokkaido, Japan

\* hi-yokoo@mua.biglobe.ne.jp

**Data Availability Statement:** All relevant data are within the manuscript and its Supporting Information files.

## Abstract

This study was performed to determine the clinical significance of adenomatous polyposis coli (APC)-binding protein end-binding 1 (EB1) in hepatocellular carcinoma (HCC) and to characterize its biochemical role in comparison with previous reports. We performed immunohistochemical staining to detect EB1 expression in tissues from 235 patients with HCC and investigated its correlations with clinicopathological features and prognosis. We also investigated the roles of EB1 in cell proliferation, migration, and tumorigenesis *in vitro* and *in vivo* by siRNA- and CRISPR/Cas9-mediated modulation of EB1 expression in human HCC cell lines. The results showed that EB1 expression was significantly correlated with several important factors associated with tumor malignancy, including histological differentiation, portal vein invasion status, and intrahepatic metastasis. Patients with high EB1 expression in HCC tissue had poorer overall survival and higher recurrence rates than patients with low EB1 expression. EB1 knockdown and knockout in HCC cells reduced cell proliferation, migration, and invasion *in vitro* and inhibited tumor growth *in vivo*. Further, genes encoding Dlk1, HAMP, and SLCO1B3 that were differentially expressed in association with EB1 were identified using RNA microarray analysis. In conclusion, elevated expression of EB1 promotes tumor growth and metastasis of HCC. EB1 may serve as a new biomarker for HCC, and genes coexpressed with EB1 may represent potential targets for therapy.

## Introduction

Liver cancer is the fourth leading cause of cancer-related deaths in the world, and hepatocellular carcinoma (HCC) is the most common form of liver cancer [1]. HCC is known to have a high relapse rate and to be characterized by high vascular invasiveness, which accounts for poor overall patient survival [2, 3]. Fujii et al. previously investigated new biomarkers for HCC by proteomics analysis, and showed that the expression level of end-binding 1 (EB1), an adenomatous polyposis coli (APC)-binding protein, was higher in hepatoma cell lines compared with normal hepatocyte cell lines [4]. Orimo et al. confirmed the same result in HCC tissues

**Funding:** This study was supported in part by JSPS KAKENHI Grant Number JP25430134 and JP16k10561, JP17k10683, and by AMED under Grant Numbers JP18fk0210041 and JP18fk0310111, and by Grant from Mitsui Life Social Welfare Foundation (2018). The funders had no role in study design, data collection and analysis, decision to publish, or preparation of the munuscript.

**Competing interests:** The authors have declared that no competing interest exist.

compared to normal liver tissues and reported that the overexpression of EB1 in HCC tissues correlated with poor prognosis [5]. Interestingly, the correlation between poor prognosis and EB1 overexpression in tumors applies to other cancers, such as esophageal cell carcinoma, breast cancer, gastric adenocarcinoma, colorectal cancer, and glioblastoma [6–10].

EB1 was originally identified by Su et al. as a protein that interacts with the tumor suppressor protein APC [11]. EB1 is now recognized to be an evolutionarily conserved microtubule tip-associated protein that localizes at the growing plus ends of microtubules and at the centrosome [12, 13]. Therefore, as an important regulator of microtubule dynamics, EB1 is critically involved in many microtubule-mediated cell activities, such as establishment and maintenance of cell polarity, localization and capture of chromosomes during mitosis, and positioning of the mitotic spindle during asymmetric cell division [13–15, 23]. However, little is known about the mechanism of EB1 involvement in HCC development.

Here we investigated the role of EB1 in HCC and its potential utility as a predictive prognostic marker for HCC patients. We further investigated the importance of EB1 expression for the proliferation, migration, and invasion of HCC cell lines. Moreover, we analyzed genes coordinately expressed with EB1 to identify new candidate biomarkers or potential targets of cancer therapy.

## Materials and methods

### Patient samples

We prepared HCC tissue samples from 235 HCC patients on whom we performed curative surgery at Hokkaido University Hospital between January 1997 and December 2006. The clinicopathological features of the patients are presented in Table 1. None of the patients received preoperative therapy. The median observation period was 5.40 years (range, 0.04–16.07 years). Tumors were classified according to the World Health Organization and International Union Against Cancer tumor-node-metastasis (TNM) classification systems [16]. Written informed consent was obtained from all patients prior to their enrollment in the study, and the study design and protocol were approved by the Institutional Review Board of Hokkaido University Hospital, Sapporo, Japan (Clinical Research Approval Number 013–0071).

### Immunohistochemical staining

We used an EnVision+ System-HRP Kit (Dako Japan, Tokyo, Japan) to perform immunohistochemistry following the manufacturer's instructions. Detailed procedures are as reported previously [17]. As a primary antibody, we used a polyclonal rabbit anti-human EB1 antibody (1:200; Santa Cruz Biotechnology, Santa Cruz, CA, USA; SC-15347). As reported previously, we selected the bile duct epithelium as an internal control for positive staining [5]. Two independent observers measured the percentage of EB1-positive tumor cells in a blinded fashion and categorized the results into groups ranging (increments of 10%) from 0% to 100%. Based on the average percentage EB1-positivity, if >30% of the HCC cells were stained, we defined the tumor as EB1-positive HCC.

### Cells

The human HCC cell lines HuH7, HepG2, JHH4, HLF, HLE, and PLC/PRF/5 were obtained from the Japanese Collection of Research Bioresources Cell Bank (JCRB, Osaka, Japan). Li-7 were obtained from RIKEN BioResource Research Center (RIKEN BRC, Tsukuba, Japan). Hep3B were purchased from American Type Culture Collection (ATCC, Rockville, MD,

**Table 1. Comparative analysis of EB1 expression and clinicopathological characteristics in patients with HCC.**

| | *n* | EB1 expression | | | | *P-value* |
|---|---|---|---|---|---|---|
| | | Negative (%) | | Positive (%) | | |
| Total | 235 | 211 | | 24 | | |
| Sex | | | | | | |
| Male | 195 | 172 | (81.5) | 23 | (95.8) | 0.0891 |
| Female | 40 | 39 | (18.5) | 1 | (4.2) | |
| Age (years) | | 61.7 | ± 9.0 | 56.3 | ± 7.6 | 0.0054 |
| Viral infection status | | | | | | |
| HBV[a] | 89 | 77 | (36.5) | 12 | (50.0) | 0.3461 |
| HCV[b] | 86 | 78 | (37.0) | 8 | (33.3) | |
| Both | 6 | 5 | (2.4) | 1 | (4.2) | |
| None | 54 | 51 | (24.1) | 3 | (12.5) | |
| Child–Pugh classification | | | | | | |
| A | 228 | 207 | (98.1) | 21 | (87.5) | 0.0251 |
| B | 7 | 4 | (1.9) | 3 | (12.5) | |
| Liver cirrhosis | | | | | | |
| Presence | 80 | 70 | (33.1) | 10 | (41.6) | 0.4957 |
| Absence | 155 | 141 | (66.9) | 14 | (58.4) | |
| AFP[c] (ng/ml) | | | | | | |
| ≥20 | 119 | 101 | (47.8) | 18 | (75.0) | 0.0171 |
| <20 | 116 | 110 | (52.2) | 6 | (25.0) | |
| PIVKA-II[d] (AU[e]/ml) | | | | | | |
| ≥40 | 137 | 118 | (55.9) | 19 | (79.1) | 0.0736 |
| <40 | 98 | 93 | (44.1) | 5 | (20.9) | |
| TNM stage | | | | | | |
| I or II | 198 | 188 | (89.1) | 10 | (41.6) | <0.0001 |
| III or IV | 37 | 23 | (10.9) | 14 | (58.4) | |
| Tumor number | | | | | | |
| Single | 172 | 160 | (75.8) | 12 | (50.0) | 0.0130 |
| Multiple | 63 | 51 | (24.1) | 12 | (50.0) | |
| Tumor size (cm) | | 4.7 | ± 3.4 | 7.4 | ± 4.8 | 0.0006 |
| Differentiation | | | | | | |
| Well or moderate | 188 | 178 | (84.3) | 10 | (41.6) | <0.0001 |
| Poor | 47 | 33 | (15.7) | 14 | (58.4) | |
| Portal vein invasion | | | | | | |
| Presence | 43 | 29 | (13.7) | 14 | (58.4) | <0.0001 |
| Absence | 192 | 182 | (86.3) | 10 | (41.6) | |
| Intrahepatic metastasis | | | | | | |
| Presence | 57 | 46 | (21.8) | 11 | (45.8) | 0.0207 |
| Absence | 178 | 165 | (78.2) | 13 | (54.2) | |

[a]HBV, hepatitis B virus

[b]HCV, hepatitis C virus

[c]AFP, α-fetoprotein

[d]PIVKA-II, protein induced by vitamin K absence; TNM, tumor-node-metastasis

[e]AU, arbitrary units.

USA). All cell lines were maintained in DMEM (Nacalai Tesque, Kyoto, Japan) supplemented with 10% FBS at 37°C in a humidified 5% $CO_2$ atmosphere.

## siRNA transfection

Control and EB1-targeting siRNAs were obtained from Invitrogen (Carlsbad, CA, USA; HSS117899 and HSS177032). The sequences of the EB1-targeting siRNAs were: 5′-CCGAAG AAACCUCUCACUUCUAGCA-3′ (siEB1-1) and 5′-GGAUCAAUGAGUCUCUGCAGUUGAA-3′ (siEB1-2). The cells were transfected with 10 nM of each siRNA. After 48 h of incubation, the efficiency of protein knockdown was confirmed by western blot analysis and quantitative real-time PCR. The cells were then used for experiments.

## Generation of EB1-knockout (KO) cell lines

EB1-KO cell lines were generated by following the procedure reported by Fukuhara et al [18]. Plasmid pX330 [19], encoding hCas9 and single-guide RNA, was obtained from Addgene (plasmid #42230). The fragments of single-guide RNA targeting the EB1 gene were inserted into the Bbs1 site of pX330 to generate pX330-EB1. HuH7 cells were transfected with pX330-EB1, and clones were established by the single-cell isolation technique. To screen for EB1-KO clones, mutations in the target loci were determined by a surveyor assay. Deficiency of protein expression was confirmed by western blot analysis.

## Re-expression of EB1 in EB1-KO cell lines

Full-length EB1 was amplified by PCR, and the products were digested and ligated into the Lenti-X vector (Takara, Tokyo, Japan). Empty vector or Lenti-X EB1 vector was co-transfected into 293T cells (Takara) to produce control or recombinant EB1-expressing lentiviruses, respectively. The lentiviral particles were harvested and used to infect EB1-KO cell lines. After several days, the infected cells were sorted, and expression of EB1 was confirmed by western blot analysis.

## Quantitative real-time PCR

Quantitative real-time PCR was carried out using SYBR Fast qPCR Mix (Takara Bio) and the primers listed in S1 Table. The specificity of the PCR products was confirmed after each amplification by a melting curve analysis, and the data were analyzed with LightCycler software (Roche, Basel, Switzerland). The target mRNA levels in each sample were normalized to β-actin mRNA.

## Western blot analysis

Cells were lysed in NP40 cell lysis buffer containing protease inhibitors. Proteins were separated by SDS-PAGE and transferred to PVDF membranes. The membranes were incubated with polyclonal rabbit anti-human EB1 (Santa Cruz Biotechnology; sc-15347) or monoclonal rabbit anti-human β-actin (Cell Signaling Technology, Danvers, MA, USA; #4970) primary antibodies and then with a horseradish-peroxidase-conjugated anti-rabbit secondary antibody (Cell Signaling Technology). Densitometric analysis of western blots was performed using a ChemiDoc XRS Plus system with Image Lab Software (Bio-Rad, Hercules, CA, USA).

## Cell proliferation assay

Cells were seeded at a density of $1 \times 10^3$ cells/well in 96-well plates and incubated for 1–4 days. A CellTiter 96 AQueous One Solution Cell Proliferation Assay Kit (Promega Corporation,

Mannheim, Germany) was used to determine viable cell numbers on days 1–4 according to the manufacturer's instructions.

## Migration and invasion assays

Migration and invasion assays were performed by placing cells into the upper chambers of a Transwell plate (BD Biosciences) without or with 100 μg/cm$^2$ Matrigel coating. Cells were added in serum-free medium, and medium supplemented with 10% FBS was added to the lower chamber as a chemoattractant. After incubation for 22 h, cells remaining on the upper side of the membrane were removed with a cotton swab, while cells adhering to the lower side were stained with Diff-Quik (Sysmex, Kobe, Japan) and visualized by light microscopy. The numbers of cells in five random fields (original magnification, ×200) were recorded.

## Animal studies

We purchased four-week-old female BALB/c nu/nu mice from CLEA Japan (Tokyo, Japan). We kept the mice under specific pathogen-free conditions in laminar-flow hoods during the experiments as previously reported [17]. All procedures involving animals and their care were approved by the Institutional Animal Care and Use Committee of National University Corporation Hokkaido University and were conducted under National University Corporation Hokkaido University Regulations on Animal Experimentation. To detect the effect of EB1 on HCC tumor growth, we mixed $1 \times 10^7$ EB1-KO HuH7 cells infected with EB1-expressing or empty-vector lentiviruses with 50 μl PBS and 50 μl Matrigel (BD Biosciences) and subcutaneously injected the mixture into the flanks of each mouse ($n$ = 5; EB1-re-expressing EB1-KO HuH7 cells in the right flank and control cells in the left flank). The sizes of the subcutaneous tumors were measured every week. After 4 weeks, the mice were euthanized by isoflurane inhalation followed by cervical dislocation and the tumors were excised and weighed. Tumor volumes (in mm$^3$) were calculated as: (shorter diameter)$^2$ × (longer diameter) × 0.5.

## Microarray analysis

Total RNA was extracted from EB1-KO HuH7 cells infected with EB1-expressing or empty-vector lentiviruses using a QIAamp RNA Blood Mini Kit (Qiagen). After properly converting the extracted RNA to the cRNA labeled with Cy3 as recommended by the manufacturers, 0.6 μg of the cRNA was fragmented and hybridized at 65˚C for 17 h to an Agilent SurePrint G3 Human GE v3 8x60K Microarray (Design ID: 072363) containing a total of 58,201 probes excluding control probes. For data analysis, the intensity values of each scanned feature were quantified using Agilent Feature Extraction software version 11.5.1.1. Normalization was performed using Agilent GeneSpring software version 13.1.1 (per chip: normalization to 75 percentile shift). Differences in transcript levels were quantified by the comparative method. We defined a ≥2.0-fold change in signal intensity as a significant difference in gene expression in this study.

## Statistical analyses

We used JMP Pro Windows version 11 (SAS Institute, Cary, NC, USA) for all statistical processing. Data are presented as mean ± SD. Comparisons of continuous data between two groups were performed using Student's unpaired $t$-test. The correlations between EB1 expression and clinicopathological features were evaluated by Fisher's exact test for categorical variables and the Mann–Whitney U test for continuous variables. The time to recurrence and overall survival were calculated from the date of first resection of the primary tumor to the

date of first radiological evidence of recurrence or death, respectively. All time-to-event end-points were computed by the Kaplan–Meier method. Patients who died without recurrence were censored in the evaluation of recurrence. Potential prognostic factors were identified by univariate analysis using the log-rank test. *P* values lower than 0.05 were considered statistically significant.

## Results

### Expression of EB1 is associated with poor prognosis of patients with HCC

First, we investigated whether EB1 expression in HCC tissue samples was correlated with the clinicopathological features and survival of our patients. The cytoplasm of tumor cells, bile duct epithelial cells, and inflammatory cells showed immunohistochemical staining of EB1, whereas no immunostaining was observed in hepatocytes in normal liver tissue areas (Fig 1A–1D). The correlations between EB1 expression and clinicopathological data are shown in Table 1. In 235 samples, positive EB1 expression (>30% of cells stained) was significantly correlated with histological differentiation ($P < 0.0001$), serum α-fetoprotein level ($P = 0.01$), serum protein induced by vitamin K absence level ($P < 0.02$), TNM stage ($P < 0.0001$), tumor size ($P = 0.001$), portal vein invasion status ($P < 0.0001$), and intrahepatic metastasis status ($P = 0.009$) (Table 1). Moreover, we investigated the correlations between EB1 expression in HCC tissue and prognostic data on HCC patients. As shown in Fig 1E and 1F, patients who had EB1-positive HCC had significantly worse prognosis than those who had EB1-negative, in regard to both cumulative recurrence rate ($P < 0.0001$) and overall survival rate ($P < 0.0001$; the median survival times of patients with EB1-positive and negative HCC were 16.5 and 3.9 years, respectively). These data suggest that EB1 is associated with tumor growth and invasion and may serve as an important predictive factor for disease recurrence and prognosis in patients with HCC.

### Knockdown of EB1 by siRNA decreases the proliferation, migration, and invasion of HCC cell lines

We examined the expression of EB1 mRNA and protein in eight human HCC cell lines (Fig 2A), and selected HLF, HLE, and HuH7 for further experiments. Efficient knockdown of EB1 by siEB1-1 and siEB1-2 in these cell lines was confirmed by quantitative real-time PCR and western blot analysis (Fig 2B). To assess the role of EB1 in tumor growth, we measured cell proliferation. Knockdown of EB1 in HLF, HLE, and HuH7 significantly decreased cell proliferation compared with control cells ($P < 0.05$, Fig 2C). Next, we assessed tumor metastasis and invasion potential using Transwell chamber migration and Matrigel invasion assays, respectively. Similar to the effect on proliferation, EB1 downregulation significantly decreased the ability of HCC cells to migrate and invade compared with control cells ($P < 0.05$, Fig 2D and 2E). These results indicate that EB1 regulates the proliferation, migration, and invasion of HCC cell lines.

### Re-expression of EB1 in EB1-KO HCC cells increases proliferation, migration, and invasion *in vitro*

To confirm these observations, we generated two EB1-KO HuH7 cell lines using CRISPR/Cas9 genome editing, and infected the cells with control or EB1-expressing lentiviruses. Efficient KO and re-expression of EB1 in the four cell lines were verified by western blot analysis (Fig 3A). The cells were then analyzed in proliferation, migration, and invasion assays. We found that re-expression of EB1 in both EB1-KO HuH7 cell lines significantly increased cell proliferation, migration, and invasion compared with control cells ($P < 0.05$, Fig 3B–3D).

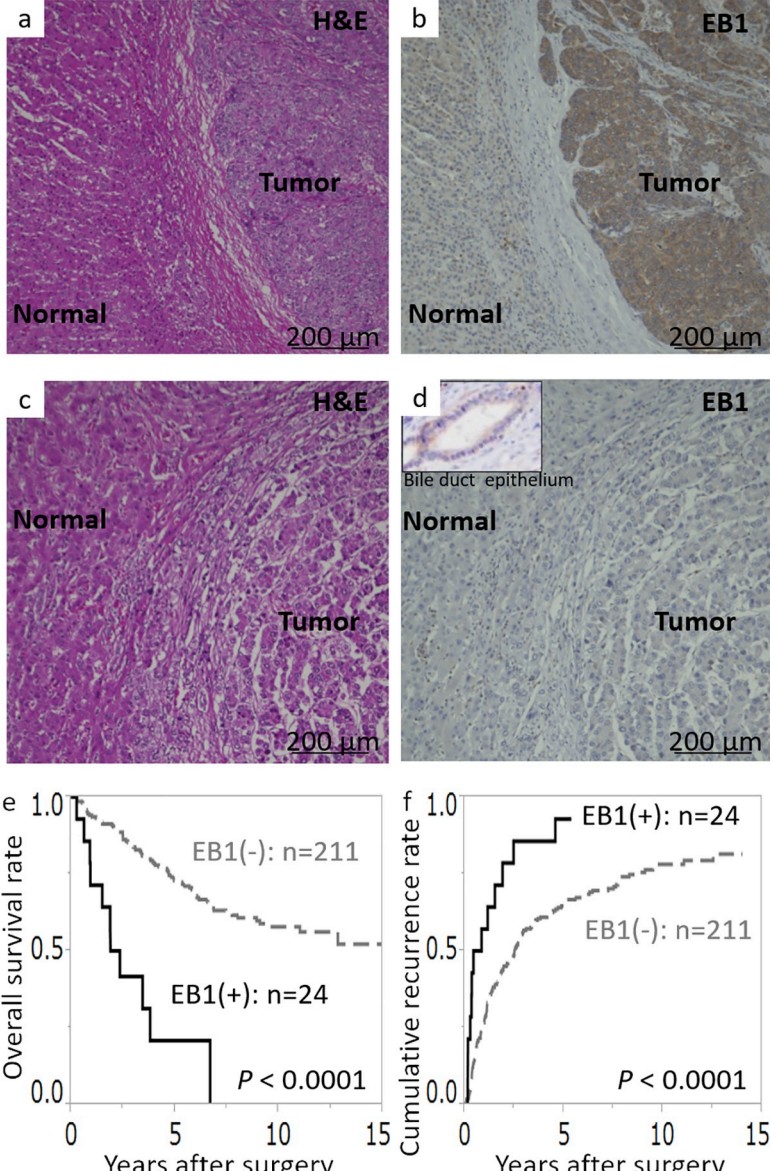

**Fig 1. Representative images of EB1 expression in HCC tissues examined by immunohistochemistry and Kaplan–Meier analysis of survival in patients with HCC.** Immunohistochemical staining of EB1 was observed in the cytoplasm of tumor cells, inflammatory cells, and bile duct epithelial cells. If more than 30% of tumor cells were more strongly stained compared with bile duct epithelial cells, the tumor was considered EB1-positive. The upper panels (a and b) show an example of EB1-positive tissue, and the lower panels (c and d) show an example of EB1-negative tissue. Kaplan–Meier survival curves for overall survival rate (e) and cumulative recurrence rate (f) according to levels of EB1 expression are shown. Solid line, patients with high EB1 expression; dotted line, patients with low EB1 expression.

These results demonstrate that EB1 plays a crucial role in malignancy-related behavior of HCC cell lines.

## Re-expression of EB1 in EB1-KO HCC cells promotes tumor growth *in vivo*

We further investigated the effect of EB1 re-expression on the growth of EB1-KO HuH7 cells injected subcutaneously into nude mice. EB1-KO HuH7 cells expressing empty vector were injected into the contralateral flank of each mouse as a control. After 4 weeks, the weight (EB1

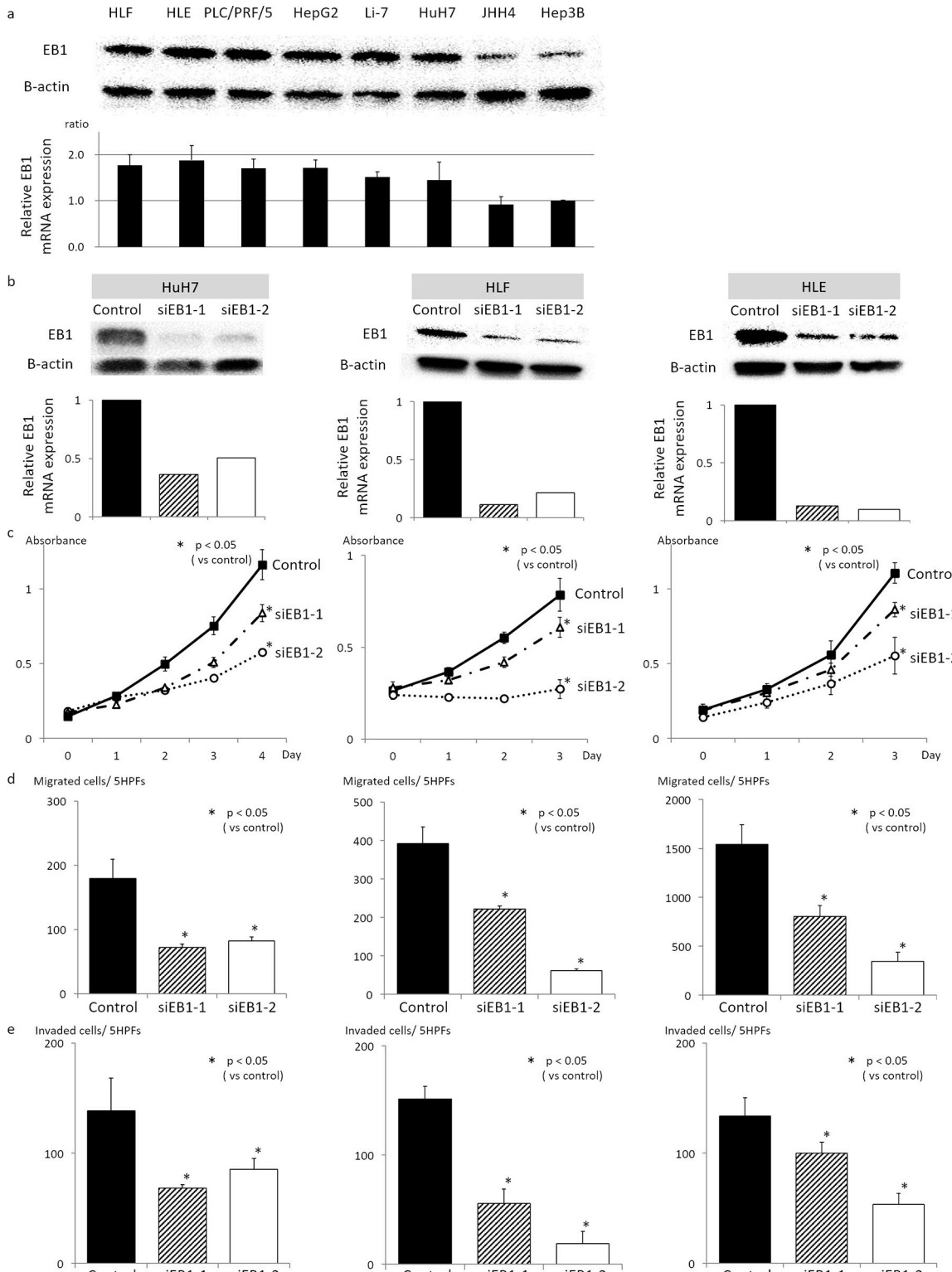

**Fig 2. Knockdown of EB1 by siRNA decreases the proliferation, migration, and invasion of HCC cell lines.** (a) Expression of EB1 mRNA and protein in HCC cell lines. (b) Real-time PCR and western blot analyses of EB1 expression in HCC cell lines with knockdown of EB1 by siRNAs. Efficient knockdown of EB1 by siEB1-1 and siEB1-2 in HuH7, HLF, and HLE was confirmed. (c) Knockdown of EB1 in HuH7, HLF, and HLE significantly decreased cell proliferation compared with control cells. (d) and (e) EB1 downregulation in HuH7, HLF, and HLE significantly decreased the ability of HCC cells to migrate (d) and invade (e) compared with control cells. The data are presented as the mean ± SD of three independent experiments in triplicate. *$P < 0.05$.

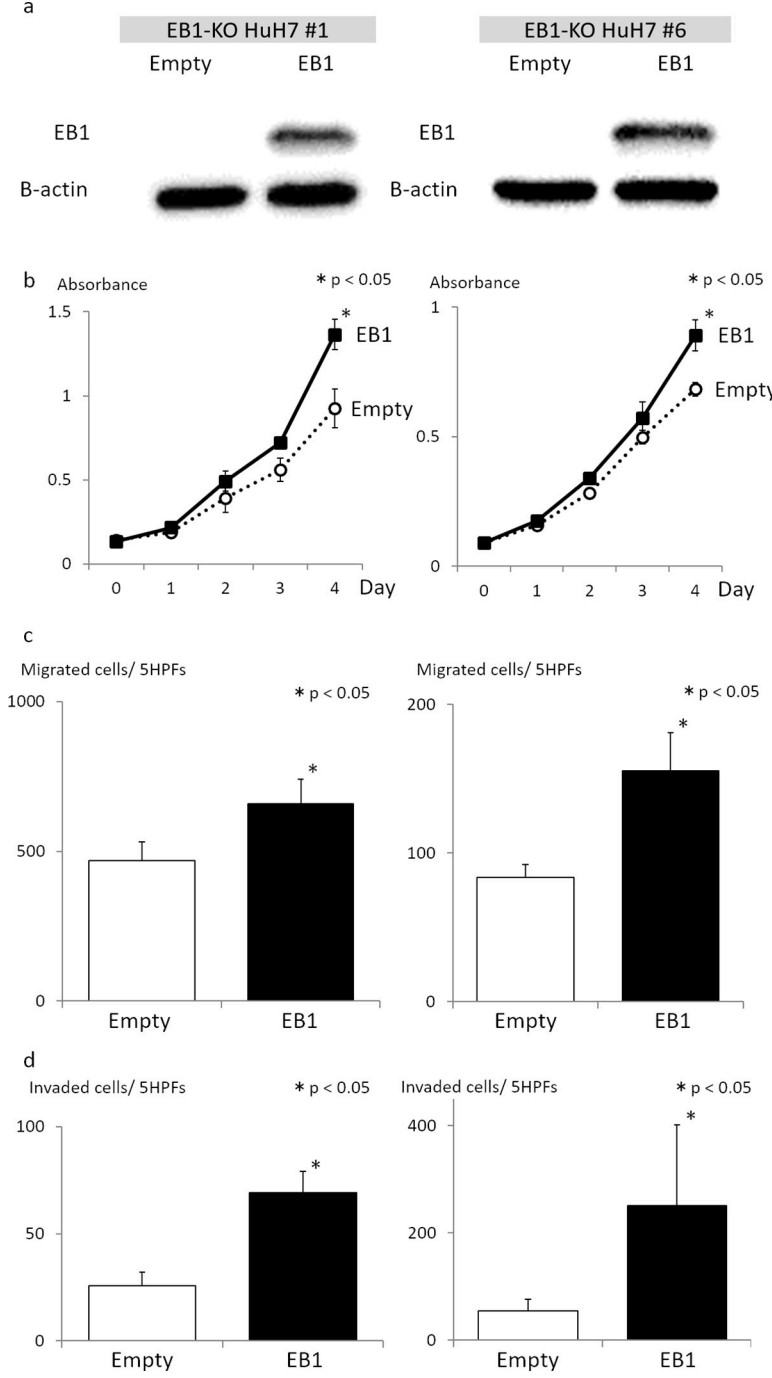

**Fig 3. Re-expression of EB1 in EB1-KO HCC cells increases proliferation, migration, and invasion *in vitro*.** (a) Western blot analysis of EB1 expression in EB1-KO HCC cells infected with empty vector-expressing or EB1-expressing lentiviruses. Efficient KO and re-expression of EB1 in the EB1-KO HuH7 cell lines were verified. (b–d) Re-expression of EB1 in both EB1-KO HuH7 cell lines significantly increased cell proliferation (b), migration (c), and invasion (d) compared with control cells. The data are presented as the mean ± SD of three independent experiments in triplicate. $^{*}P < 0.05$.

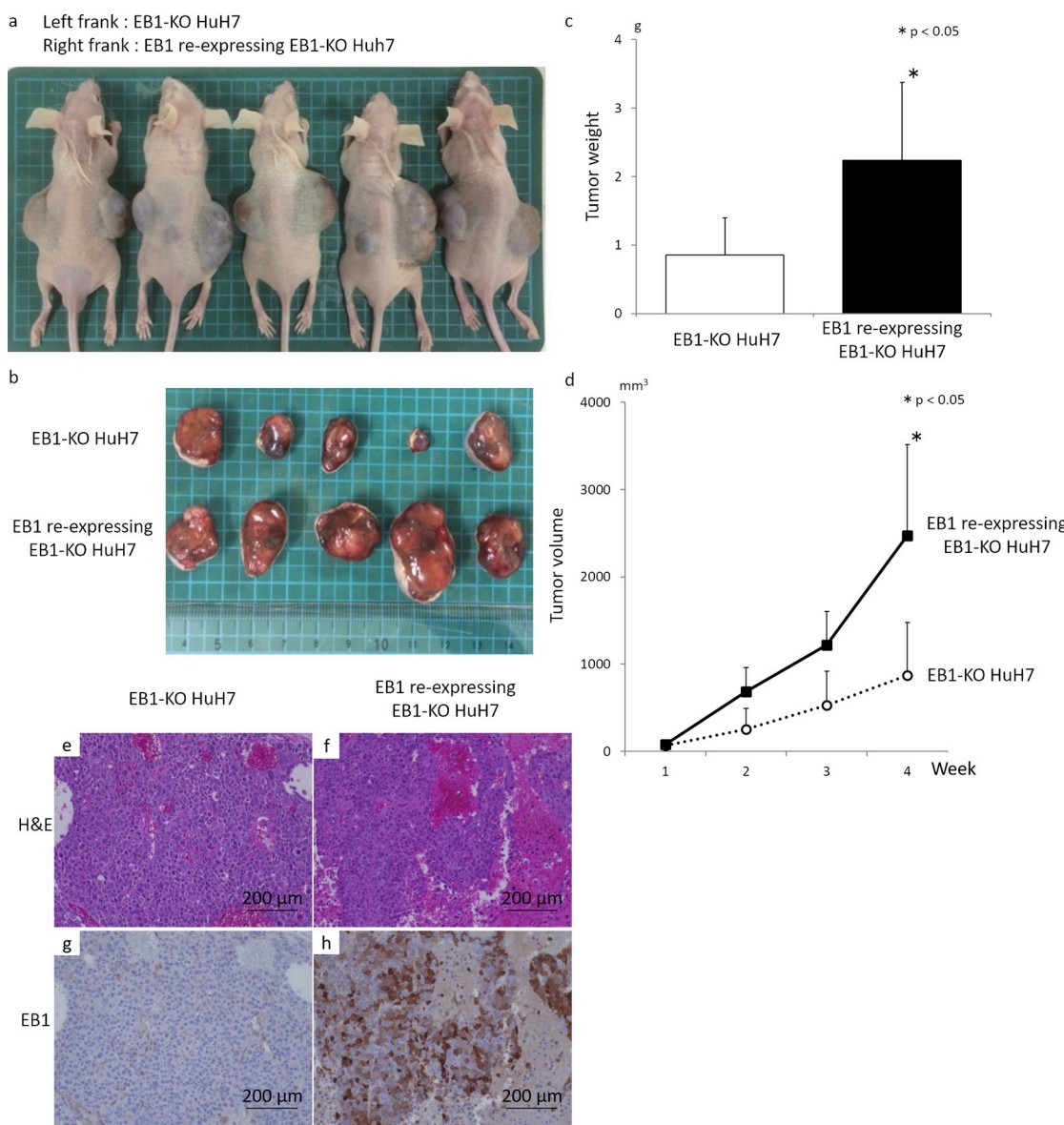

**Fig 4. Re-expression of EB1 in EB1-KO HCC cells promotes tumor growth *in vivo*.** (a) Photographs of mice at 4 weeks after injection with EB1-KO HuH7 cells infected with EB1-expressing lentivirus in the right flank (EB1) and EB1-KO HuH7 cells infected with empty vector-expressing lentivirus in the left flank (Empty) as a control. (b) Photographs of excised tumors at 4 weeks. (c) and (d) The weight (c) and volume (d) of tumors with EB1 re-expression were higher than those of control tumors. (e-h) Representative images of EB1 expression in excised tumors examined by immunohistochemistry. Immunohistochemical staining of EB1 in the excised tumors confirmed the depletion and re-expression of EB1 in the cells carrying the empty vector (e and g) and EB1 expression vector (f and h), respectively. $^{*}P < 0.05$.

re-expressing EB1-KO HuH7 cells, 2.23 ± 1.13 g; EB1-KO HuH7 cells, 0.85 ± 0.54 g; $P < 0.05$, Fig 4C) and volume (EB1 re-expressing EB1-KO HuH7 cells, 2474 ± 1044 mm$^3$; EB1-KO HuH7 cells, 872 ± 605 mm$^3$, $P < 0.05$, Fig 4A, 4B, and 4D) of tumors with EB1 re-expression were higher than those of control tumors. Immunohistochemical staining of EB1 in the excised tumors confirmed the depletion and re-expression of EB1 in the cells carrying the empty vector and EB1 expression vector, respectively (Fig 4E–4H). These results demonstrate that EB1 plays an important role in tumor growth *in vivo*.

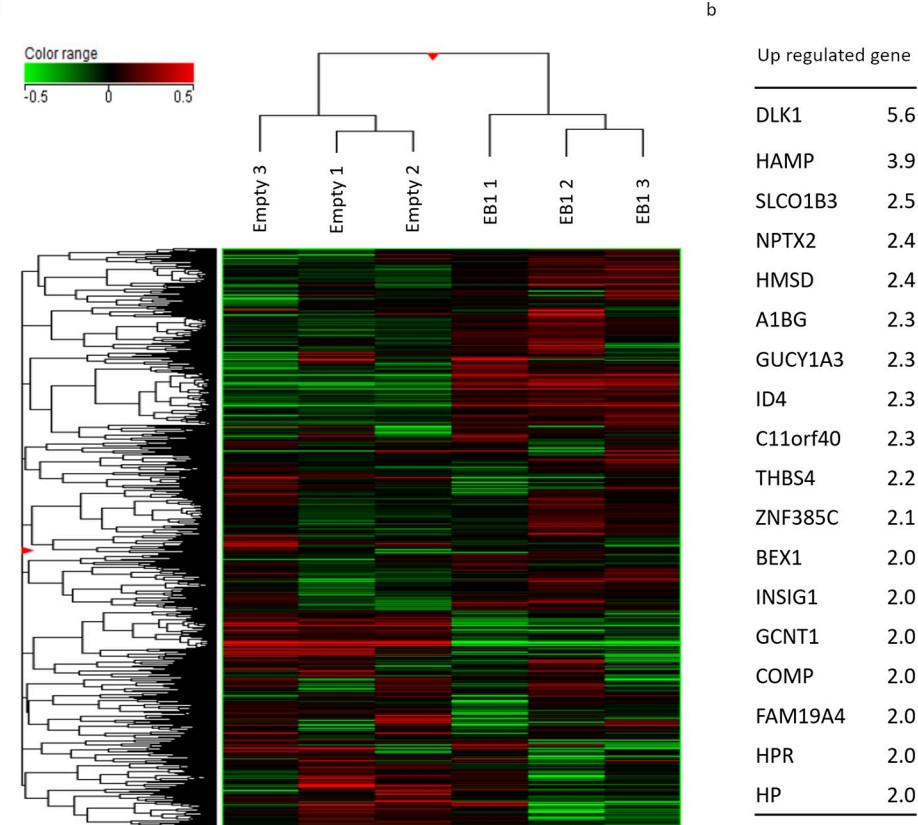

**Fig 5. Genes expressed in association with upregulated EB1 expression.** (a and b) RNA microarray analysis to identify genes differentially expressed in the presence of EB1. Comparison of gene expression levels in EB1-KO HuH7 cells infected with control and EB1-expressing lentiviruses identified a number of genes upregulated by EB1 re-expression.

## Genes expressed in association with upregulated EB1 expression

Finally, we performed an RNA microarray analysis (Agilent SurePrint G3 Human GE v3 8x60K Microarray, Design ID: 072363) to identify genes differentially expressed in the presence of EB1. Comparison of gene expression in EB1-KO HuH7 cells infected with control and EB1-expressing lentiviruses identified a number of genes upregulated upon EB1 re-expression (Fig 5A). Among them, Delta-like 1 homolog (Dlk1) was the most significantly upregulated, followed in descending order by Hepcidin antimicrobial peptide (HAMP), and Solute carrier organic anion transporter family member 1B3 (SLCO1B3) (Fig 5B). We suggest that upregulation of EB1 expression correlates with the expression of these genes.

## Discussion

In the present study, we showed that EB1 expression is an important predictive marker for survival and disease recurrence in patients with HCC. EB1 overexpression was significantly correlated with poor histological differentiation, advanced TNM stage, large tumor size, advanced portal vein invasion status, and high intrahepatic metastasis status. *In vitro* studies revealed that overexpression of EB1 promoted HCC cell proliferation, motility, and invasion. Our *in vivo* study validated the *in vitro* findings by demonstrating that expression of EB1 promoted the growth of HCC tumors. Moreover, our microarray analysis indicated that EB1 expression

might correlate with the expression of genes such as those encoding Dlk1, HAMP, and SLCO1B3 (Fig 5B). Our findings thus suggest that EB1 plays an important role in the growth, migration, and invasion of HCCs and that these genes may be involved in the mechanism of EB1-associated tumor development.

High expression of EB1 has been reported in several malignancies to date, including esophageal squamous cell carcinoma, glioblastoma, colon cancer, gastric carcinoma, breast cancer, and HCC [4–10]. In particular, EB1 expression was associated with histological grade, pathological TNM stage, and lymph node metastasis in breast cancer [7] and with serum α-fetoprotein level, TNM stage, tumor number and size, histological differentiation, portal vein invasion status, and intrahepatic metastasis in HCC [5]. Our data are consistent with these reports and strongly suggest that EB1 expression is correlated with tumor growth and invasiveness. Moreover, EB1 was reported to be a significant independent predictor of overall survival in patients with glioblastoma, colon cancer, and HCC [5, 8–10]. In this study, we validated that EB1 was an important predictor of prognosis and recurrence in our cohort of patients with HCC. These observations strongly suggest that EB1 is involved in the progression of HCC.

EB1 was originally identified in 1995 by Su et al. [11] who used a yeast two-hybrid screen to identify proteins that bind to the tumor suppressor APC. Since APC regulates the canonical Wnt signaling pathway [20, 21], we hypothesized that EB1 may be a component of this pathway. However, using a β-catenin/Tcf luciferase reporter system [22], we found no evidence that EB1 promotes Wnt3a-induced activation of Wnt/β-catenin signaling (S1 Fig), in contrast to a previous report [6].

A recent study showed that EB1 can promote Aurora-B kinase activity by preventing its inactivation by protein phosphatase 2A [23]. However, when we examined this in our HuH7 EB1-KO cell system, we did not observe increased Aurora-B kinase activity in EB1-KO cells after re-expression of EB1 (S2 Fig).

In addition, our data revealed that EB1 promoted not only cell proliferation, but also cell migration and invasion. These are complex processes requiring coordinated re-organization of the actin and microtubule cytoskeletons under both physiological and pathological conditions, including angiogenesis and tumor cell metastasis [24–28]. Filopodia, lamellipodia, and invadopodia are protrusive structures localized at the cell front during mobility [24] and invasion [25]. Microtubules cooperate with the actin cytoskeleton to elongate invadopodia during invasion [25] and to maintain cell polarization at the leading and trailing edges during cell migration [26–28]. These microtubule functions are controlled or mediated by microtubule plus-end tracking proteins, including EB1 [28]. Indeed, knockdown or overexpression of EB1 decreased or increased cell migration or invasion, respectively, in glioblastoma cell lines [10], colon cancer cell lines [9], melanoma cell lines [29], and our HCC cell lines. Therefore, we suggest that EB1 is necessary to maintain cell polarity and direction during migration or invasion of cancer cells.

However, these observations leave us with the unanswered question of how EB1 is involved in the development and progression of HCC. Our microarray analysis revealed that EB1 overexpression might be correlated with upregulation of the expression of genes encoding Dlk1, HAMP, and SLCO1B3. Dlk1 is an imprinted gene and is only transcribed from the paternal allele in humans [30, 31]. Intriguingly, previous work showed that Dlk1 plays a vital role in liver development and HCC oncogenesis [30–36]. Moreover, Xu et al. [37] recently reported that Dlk1-positive HCC cells show more robust proliferation and tumorigenicity compared with Dlk1-negative cells. The HAMP gene encodes hepcidin, which is primarily produced in the liver and is a key iron-regulatory hormone [38]. The levels of serum hepcidin in patients with non-small cell lung cancer, breast cancer, and pancreatic cancer are significantly higher compared with those of healthy individuals or those with benign disease [39–41]. Moreover,

high levels of serum or cellular hepcidin in patients with non-small cell lung cancer or pancreatic cancer is associated with metastasis, TNM stage, and poor prognosis [39, 41]. The gene encoding SLCO1B3 is a member of the SLCO gene superfamily, which encodes organic anion transporting polypeptide (OATP). OATP1B1 and OATP1B3 deficiency causes human Rotor syndrome by interrupting the reuptake of conjugated bilirubin reuptake into the liver [42]; however, high levels of OATP1B3 transcripts are significantly associated with colorectal cancer and prostate cancer, suggesting that OATP1B3 represents a promising biomarker for those cancers [43, 44]. Therefore, we suggest that by upregulating the expression of these genes, EB1 promotes cell proliferation, tumor growth, and metastasis of HCC.

There are several limitations to our study. First, there are some inconsistency of immunohistochemistry data compared with those of Orimo et al [5]. This may be explained by the differences of the immunohistochemical staining method and the differences in patients' backgrounds as we conducted a retrospective analysis. Further investigation is required to improve accuracy and reproducibility in the assessment of EB1 expression in the clinical setting. Second, although the p values of the results of our experiments indicate significant differences, statistical power may be insufficient because of the limited number of samples. Nevertheless, our data are consistent with findings of others that elevated EB1 expression significantly correlates with the proliferation, migration, and invasion of cancer cells [9, 10, 29]. Third, the mechanism by which EB1 regulates the expression of the above candidate genes is unclear. For example, Sun et al. [21] reported that EB1 binds to and regulates the activity of Aurora-B kinase. Thus, EB1 may regulate the expression of the candidate genes by binding to a protein that regulates their expression. However, further investigation is required to clarify this issue.

In conclusion, our data show that EB1 expression is significantly correlated with various clinicopathological parameters, leading to a poor prognosis in patients with HCC. We further found that genes such as those encoding Dlk1, HAMP, and SLCO1B3 may be involved in the mechanism by which EB1 supports HCC cell proliferation, migration, and invasion. The present findings suggest that an unknown EB1 axis may represent a new therapeutic target for HCC.

## Supporting information

**S1 Checklist.**
(PDF)

**S1 Fig. The relation between EB1 and Wnt signaling.** SuperTopFlash 293 reporter cells containing the αβ-catenin/Tcf reporter system [19] were transfected with empty or EB1 expression pcDNA3.1 plasmid, stimulated with Wnt3a-conditioned medium, and measured for their luciferase activities. Although EB1 elevation was detected at the protein level, there was no difference in luciferase activities in the cells transfected with the empty or EB1 expression plasmid.
(TIF)

**S2 Fig. The relation between EB1 and aurora-B kinase activity.** There was no significant activation of aurora-B kinase in EB1-KO HuH7 cells re-expressing EB1 compared with EB1-KO HuH7 cells.
(TIF)

**S3 Fig.**
(TIF)

**S4 Fig.**
(TIF)

**S1 Raw images.**
(TIF)

**S1 Table. Primers.**
(DOCX)

# Acknowledgments

We thank the members of Department of Gastroenterological Surgery I and Department of Surgical Pathology, especially the laboratory assistants, Nozomi Kobayashi, Sayaka Miyoshi and Ayae Nange, for helpful technical assistance. We thank Tadasuke Tsukiyama (Department of Biochemistry, Hokkaido University Graduate School of Medicine, Sapporo, Japan) for helpful discussions and technical assistance with Wnt signaling. We thank Takasuke Fukuhara and Yoshiharu Matsuura (Department of Molecular Virology, Research Institute for Microbial Diseases, Osaka, Japan) for technical assistance with CRISPR/Cas9 technology and supporting of generation of EB1-KO cell lines. Finally, we thank Anne M. O'Rourke, PhD, and Alison Sherwin, PhD, from Edanz Group (www.edanzediting.com/ac) for editing drafts of this manuscript.

# Author Contributions

**Conceptualization:** Takeshi Aiyama, Tatsuya Orimo, Hideki Yokoo, Toshiya Kamiyama.

**Data curation:** Takeshi Aiyama, Takanori Ohata, Kanako C. Hatanaka, Yutaka Hatanaka, Moto Fukai, Toshiya Kamiyama.

**Formal analysis:** Takeshi Aiyama, Moto Fukai.

**Funding acquisition:** Takeshi Aiyama, Tatsuya Orimo, Hideki Yokoo, Moto Fukai, Toshiya Kamiyama.

**Investigation:** Takeshi Aiyama, Takanori Ohata.

**Methodology:** Takeshi Aiyama, Tatsuya Orimo, Hideki Yokoo, Kanako C. Hatanaka, Yutaka Hatanaka, Moto Fukai, Toshiya Kamiyama.

**Project administration:** Takeshi Aiyama, Tatsuya Orimo, Kanako C. Hatanaka, Yutaka Hatanaka, Moto Fukai, Toshiya Kamiyama.

**Resources:** Takeshi Aiyama, Tatsuya Orimo, Hideki Yokoo, Yutaka Hatanaka, Moto Fukai, Toshiya Kamiyama.

**Supervision:** Tatsuya Orimo, Moto Fukai, Toshiya Kamiyama, Akinobu Taketomi.

**Validation:** Takeshi Aiyama.

**Writing – original draft:** Takeshi Aiyama.

**Writing – review & editing:** Tatsuya Orimo, Hideki Yokoo, Kanako C. Hatanaka, Yutaka Hatanaka, Moto Fukai, Toshiya Kamiyama, Akinobu Taketomi.

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
