## [Decision Letter · Decision Letter 0]

12 Jun 2020

PONE-D-20-12629

Adenomatous polyposis coli-binding protein end-binding 1 promotes hepatocellular carcinoma growth and metastasis

PLOS ONE

Dear Dr. Yokoo,

Thank you for submitting your manuscript to PLOS ONE. After careful consideration, we feel that it has merit but does not fully meet PLOS ONE’s publication criteria as it currently stands. Therefore, we invite you to submit a revised version of the manuscript that addresses the points raised during the review process.

Two experts have reviewed the manuscript and foudn that the study needs to be strengthened with more clear presentatinos via more critical statistic analysis, controls, and WB or images in higher qualities.

We look forward to receiving your revised manuscript.

Kind regards,

Jung Weon Lee, Ph.D.

Academic Editor

PLOS ONE

Journal Requirements:

2. Our staff editors have determined that your manuscript is likely within the scope of our Cancer Metastasis Call for Papers. This editorial initiative is headed by a team of Guest Editors for PLOS ONE: Joe Ramos (University of Hawai'i), Shengyu Yang (Penn State University), Helen Fillmore (University of Portsmouth) and Tobias Zech (University of Liverpool). The Collection will encompass a diverse range of research articles about metastasis, including mechanisms of cell motility, invasion and the tumor microenvironment, as well as advances in the development of anti-metastatic therapies.  Additional information can be found on our announcement page: https://collections.plos.org/s/cancer-metastasis

If you would like your manuscript to be considered for this collection, please let us know in your cover letter and we will ensure that your paper is treated as if you were responding to this call.  Please note that being considered for the Collection does not require additional peer review beyond the journal’s standard process and will not delay the publication of your manuscript if it is accepted by PLOS ONE. If you would prefer to remove your manuscript from collection consideration, please specify this in the cover letter.

3. Thank you for including your ethics statement: 'All procedures involving animals and their care were approved by the Ethics Committee of Hokkaido University'  

(a)  Please state whether the provided ethics committee contains animal welfare experts or whether an animal ethics or IACUC committee reviewed and approved the study. Please provide the full name of the committee that reviewed and approved the study  

(b) Once you have amended this/these statement(s) in the Methods section of the manuscript, please add the same text to the “Ethics Statement” field of the submission form (via “Edit Submission”).

For additional information about PLOS ONE ethical requirements for human subjects research, please refer to " ext-link-type="uri" xlink:type="simple">http://journals.plos.org/plosone/s/submission-guidelines#loc-human-subjects-research."

4. Please provide additional information about each of the cell lines used in this work, including any quality control testing procedures (authentication, characterisation, and mycoplasma testing). For more information, please see http://journals.plos.org/plosone/s/submission-guidelines#loc-cell-lines.

5. To comply with PLOS ONE submissions requirements, please provide the method of euthanasia in the Methods section of your manuscript.

6. As part of your revision, please complete and submit a copy of the ARRIVE Guidelines checklist, a document that aims to improve experimental reporting and reproducibility of animal studies for purposes of post-publication data analysis and reproducibility: https://www.nc3rs.org.uk/arrive-guidelines. Please include your completed checklist as a Supporting Information file. Note that if your paper is accepted for publication, this checklist will be published as part of your article.

7. We noticed minor instances of text overlap with the following previous publication(s), which need to be addressed:

(1) https://www.pnas.org/content/105/20/7153.full

The text that needs to be addressed involves the Introduction section.

In your revision please ensure you cite all your sources (including your own works), and quote or rephrase any duplicated text outside the methods section. Further consideration is dependent on these concerns being addressed.

8. PLOS ONE now requires that authors provide the original uncropped and unadjusted images underlying all blot or gel results reported in a submission’s figures or Supporting Information files. This policy and the journal’s other requirements for blot/gel reporting and figure preparation are described in detail at https://journals.plos.org/plosone/s/figures#loc-blot-and-gel-reporting-requirements and https://journals.plos.org/plosone/s/figures#loc-preparing-figures-from-image-files. When you submit your revised manuscript, please ensure that your figures adhere fully to these guidelines and provide the original underlying images for all blot or gel data reported in your submission. See the following link for instructions on providing the original image data: https://journals.plos.org/plosone/s/figures#loc-original-images-for-blots-and-gels.

Additional Editor Comments (if provided):

Reviewers' comments:

Reviewer's Responses to Questions

**Comments to the Author**

1. Is the manuscript technically sound, and do the data support the conclusions?

Reviewer #1: Partly

Reviewer #2: Partly

2. Has the statistical analysis been performed appropriately and rigorously? 

Reviewer #1: No

Reviewer #2: Yes

3. Have the authors made all data underlying the findings in their manuscript fully available?

Reviewer #1: Yes

Reviewer #2: No

4. Is the manuscript presented in an intelligible fashion and written in standard English?

Reviewer #1: Yes

Reviewer #2: Yes

5. Review Comments to the Author

Reviewer #1: In the manuscript, the authors described a study to determine the clinical significance of adenomatous polyposis coli-binding protein end-binding 1 (EB1) in hepatocellular carcinoma (HCC). The authors performed various experiments and concluded that elevated expression of EB1 promotes tumor growth and metastasis of HCC, suggesting using EB1 as a HCC biomarker. While evidence from multiple experiments seems supportive of the conclusion, the sample sizes are small in most of the experiments, and the statistical significance of many tests seems marginal.  A few comments:

[Page 9, 10; Table 1] For numerical variables like age and tumor size, it would be good to provide standard deviation in addition to the mean values.  For categorical variables, the p-values calculated seem different from those by Fisher's exact tests.

[Page 12; Fig 2c, d, and e]

The sample size may be too small to draw meaningful conclusion, and it would be good to state the mean and standard deviation values for proliferation, migration, and invasion in the manuscript.

[Page 12; Fig 3b, c, and d]

The sample size is too small, and the p-values may not be valid.  It might be better to include more biological replicates.

[Page 13; Fig 4]

Again the sample size is too small to draw any conclusion with statistical significance.  More samples would be recommended.

[Page 14; Fig 5b]

While the list of up-regulated genes is interesting, it would be of interest to analyze any potential functional or pathway enrichment.

Reviewer #2: The manuscript by Aiyama and coworkers confirms the involvement of EB1 in HCC by showing that its expression is associated with poor survival and high recurrence rate. Additionally, in vitro and in vivo experiments show involvement of EB1 in proliferation, migration, invasion and tumor growth. The study appears solid, but there are a few issues, mainly regarding lack of information in the methods, that should be addressed.

1) Was the assesment of staining done just by eye or was imageJ or equivalent used ? Where is the 30% staining cutoff based on (Orimo 2008 study used 50%?!), and how was this exactly determined?

2) For all-in vitro experiments information is lacking regarding technical replicates and the number of independent experiments performed.

3) Is it certain the b-actin mRNA remains stable upon EB1 knockdown or overexpression ? It is standard to use 2-3 genes for qPCR normalization, not just 1.

4) Same for WB normalization, is b-actin proven to remain stable? Has normalization also been done using a whole protein stain?

5) In the migration assays, was the membrane of the transwell plate completely uncoated? And for invasion, what was the volume of matrigel used to coat the transwells?

6) Fiugre 2a: to choose HLF, HLE and HuH7 appears a bit random as all but JHH4 and HepG2 appear suitable. Please rephrase line 220.

7) Why was specifically HuH7 chosen fort he knockout experiments? Please clarify.

8) All data in figure 3 should be shown in comparison to the original HuH7 cells, especially to confirm efficient knockout.

9) Line 300 : Unclear why SLCO1B3 is discussed and for instance NPTX2 and HMSD not. Additionally, the data from the microarray should be validated by at least qPCR before these can be considered reliable enough to thoroughly discuss in line 341 to 359.

10) Line 363-364 is unclear, this statement requires further explanation.

11) Overall, the Western blot images quality is quite poor and higher resolution images are needed.

6. PLOS authors have the option to publish the peer review history of their article (what does this mean?). If published, this will include your full peer review and any attached files.

Reviewer #1: No

Reviewer #2: No

---

## [Author Response · Author response to Decision Letter 0]

27 Jul 2020

July 27, 2020:

Jung Weon Lee, Ph.D.

Academic Editor

PLOS ONE

Dear Dr. Lee,

Thank you for the opportunity to submit a revised version of our manuscript (PONE-D-20-12629) titled, “Adenomatous polyposis coli-binding protein end-binding 1 promotes hepatocellular carcinoma growth and metastasis,” for publication as a Research Article in PLOS ONE. We greatly appreciate the reviewers’ incisive comments that have helped us to significantly improve the manuscript. Our point-by-point responses to these comments are stated on the next page. And we appreciate your consideration for the collection of “Cancer Metastasis Call for Papers”. In addition, we uploaded the original uncropped and unadjusted images underlying all blot results reported in our submission’s figures. We hope that our paper is now suitable for publication in the PLOS ONE.

Sincerely,

Hideki Yokoo

Department of Gastroenterological Surgery I

Hokkaido University Graduate School of Medicine

N-14 W-5, Kita-ku, Sapporo 060-8648, Japan.

E-mail: hi-yokoo@mua.biglobe.ne.jp

Tel: +81-11-706-5927

Fax: +81-11-717-7515.

 

Reviewer(s)' Comments to Author:

Reviewer: 1

Comments and our Responses:

#1. [Page 9, 10; Table 1] For numerical variables like age and tumor size, it would be good to provide standard deviation in addition to the mean values. For categorical variables, the p-values calculated seem different from those by Fisher's exact tests.

Response: Thank you for your advice. We have added the standard deviation in our table, and corrected the p-values (Page 10, 11; Table 1).

#2. [Page 12; Fig 2c, d, and e] The sample size may be too small to draw meaningful conclusion, and it would be good to state the mean and standard deviation values for proliferation, migration, and invasion in the manuscript.

Response: Thank you for your comment. We noticed several mistakes of data analysis in migration and invasion assay and have changed the figure. In addition, We have added the following sentence in the figure legend. “The data are presented as the mean ± SD of three independent experiments in triplicate.” The mean and standard deviation values for all assays are as follows.

Proliferation assay (n=3):

HuH7 control siRNA, 1.16 ± 0.10; HuH7 EB1 siRNA1, 0.83 ± 0.05; HuH7 EB1 siRNA2, 0.57 ± 0.02.

HLF control siRNA, 0.80 ± 0.07; HLF EB1 siRNA1, 0.63 ± 0.06; HLF EB1 siRNA2 0.28 ± 0.03.

HLE control siRNA, 1.10 ± 0.04; HLE EB1 siRNA1, 0.86 ± 0.06; HLE EB1 siRNA2, 0.55 ± 0.12.

Migration assay (n=3):

HuH7 control siRNA, 179.3 ± 29.7; HuH7 EB1 siRNA1, 72.0 ± 5.2; HuH7 EB1 siRNA2, 82.0 ± 6.0 migrated cells / 5HPFs.

HLF control siRNA, 392.0 ± 43.8; HLF EB1 siRNA1, 222.0 ± 8.1; HLF EB1 siRNA2 62.0 ± 4.5 migrated cells / 5HPFs.

HLE control siRNA, 1544.3 ± 198.8; HLE EB1 siRNA1, 804.3 ± 113.3; HLE EB1 siRNA2, 347.0 ± 89.7 migrated cells / 5HPFs.

Invasion assay (n=3):

HuH7 control siRNA, 138.3 ± 29.7; HuH7 EB1 siRNA1, 68.3 ± 3.0; HuH7 EB1 siRNA2, 85.3 ± 10.0 invaded cells / 5HPFs.

HLF control siRNA, 151.3 ± 11.5; HLF EB1 siRNA1, 55.6 ± 13.2; HLF EB1 siRNA2 18.6 ± 11.2 invaded cells / 5HPFs.

HLE control siRNA, 133.6 ± 16.4; HLE EB1 siRNA1, 100.0 ± 9.84; HLE EB1 siRNA2, 53.6 ± 9.8 invaded cells / 5HPFs.

#3. [Page 12; Fig 3b, c, and d] The sample size is too small, and the p-values may not be valid. It might be better to include more biological replicates.

Response: Thank you for your comment. We have conducted a cell counting proliferation assay for EB1-KO HuH7 cells (starting with 10 million cells per well, counting in day4), and the results were consistent with the MTS proliferation assay described in our manuscript (EB-KO HuH7 #1 Empty, 91.1 ± 4.7; EB1-KO HuH7 #1 EB1 re-expressing, 178.6 ± 23.5 million cells per well (n=3). EB-KO HuH7 #6 Empty, 60.0 ± 5.7; EB1-KO HuH7 #6 EB1 re-expressing, 80.6 ± 3.5 (n=3)). However, we noticed a mistake of data analysis in migration assay and have changed the figure. In addition, We have also added the following sentence in the figure legend. “The data are presented as the mean ± SD of three independent experiments in triplicate.” The mean and standard deviation values for all assays are as follows.

Proliferation assay (n=3):

EB-KO HuH7 #1 Empty, 0.92 ± 0.11; EB1-KO HuH7 #1 EB1 re-expressing, 1.36 ± 0.09.

EB-KO HuH7 #6 Empty, 0.68 ± 0.02; EB1-KO HuH7 #6 EB1 re-expressing, 0.89 ± 0.05.

Migration assay (n=3):

EB-KO HuH7 #1 Empty, 469.0 ± 61.2; EB1-KO HuH7 #1 EB1 re-expressing, 658.3 ± 82.8 migrated cells / 5HPFs.

EB-KO HuH7 #6 Empty, 83.8 ± 8.4; EB1-KO HuH7 #6 EB1 re-expressing, 155.2 ± 25.9 migrated cells / 5HPFs.

Invasion assay (n=3):

EB-KO HuH7 #1 Empty, 25.6 ± 6.4; EB1-KO HuH7 #1 EB1 re-expressing, 69.2 ± 9.9 invaded cells / 5HPFs.

EB-KO HuH7 #6 Empty, 61.6 ± 23.1; EB1-KO HuH7 #6 EB1 re-expressing, 343.3 ± 109.1 invaded cells / 5HPFs.

#4. [Page 13; Fig 4] Again the sample size is too small to draw any conclusion with statistical significance. More samples would be recommended.

Response: Thank you for your advice. We re-evaluated the data of the animal studies. However, the p-values were under 0.05. We have added the mean ± SD of tumor weight and volume in the result section (Page 265 to 267)

Tumor weight (n=5):

EB1-re-expressing EB1-KO HuH7 cells (EB1), 2.23 ± 1.13 g; Control EB1-KO HuH7 cells (Empty), 0.85 ± 0.54 g; P 0.05.

Tumor volume (n=5):

EB1, 2474 ± 1044 mm3; Empty, 872 ± 605 mm3; P 0.05.

#5. [Page 14; Fig 5b] While the list of up-regulated genes is interesting, it would be of interest to analyze any potential functional or pathway enrichment.

Response: Thank you for an interesting proposal. We are interested in the association between Dlk-1 and EB1, and analyzing it. We can not show the data, but we know that the mRNA and the protein of Dlk-1 is increased in the EB1-re-expressing EB1-KO HuH7 cells compared with the control EB1-KO HuH7 cells. We hope to publish the results in the future.

Reviewer: 2

Comments and our Responses:

#1. Was the assesment of staining done just by eye or was imageJ or equivalent used ? Where is the 30% staining cutoff based on (Orimo 2008 study used 50%?!), and how was this exactly determined?

Response: Thank you for your comments. The assessment of immunohistochemical staining was done by eyes of two independent observers in a blinded fashion. The 30% staining cutoff value was decided by calculating the average of the EB1 positive percentage. There were 10 patients with 10% EB1 positive, 12 with 20%, 8 with 30%, 3 with 40%, 2 with 50%, 1 with 60%, 3 with 70%, 4 with 80%, 0 with 90%, 4 with 100%, so the average was 34.5%. Therefore, we analyzed our clinical data with 30% staining cutoff value. If we analyzed our data with 50% staining cutoff value, the median survival times of patients with EB1-positive and negative HCC were 15.0 and 2.1 years, respectively (P 0.0001).

#2. For all-in vitro experiments information is lacking regarding technical replicates and the number of independent experiments performed.

Response: Thank you for your pointing out. We have added the following sentence in the figure legends. “The data are presented as the mean ± SD of three independent experiments in triplicate.” (Page 12, line 241 and 242; Page 13, line 258 and 259)

#3. Is it certain the b-actin mRNA remains stable upon EB1 knockdown or overexpression ? It is standard to use 2-3 genes for qPCR normalization, not just 1.

Response: Thank you for your advice. We have checked that the b-actin mRNA remains stable upon EB1 knockdown with siRNAs transfection comparing with GAPDH mRNA. The data is uploaded as a supplementary figure S3.

#4. Same for WB normalization, is b-actin proven to remain stable? Has normalization also been done using a whole protein stain?

Response: Thank you for your advice. We have checked that the b-actin protein remains stable upon EB1 knockout and re-expressed. The data is also uploaded as a supplementary figure S3.

#5. In the migration assays, was the membrane of the transwell plate completely uncoated? And for invasion, what was the volume of matrigel used to coat the transwells?

Response: Thank you for your questions. In the migration assay, the membrane of the transwell plate was completely uncoated. And for invasion, the volume of matrigel used to coat the transwell was 100 µg/cm2 according to manufacturer’s catalogue.

#6. Fiugre 2a: to choose HLF, HLE and HuH7 appears a bit random as all but JHH4 and HepG2 appear suitable. Please rephrase line 220.

Response: Thank you for your comment. We have rephrased it. (Page 11, line 223)

#7. Why was specifically HuH7 chosen for the knockout experiments? Please clarify.

Response: Thank you for your comment. As the generation of EB1-KO cell line using the CRISPR/Cas9 technology was supported by Takasuke Fukuhara and Yoshiharu Matsuura (Department of Molecular Virology, Research Institute for Microbial Diseases, Osaka, Japan), they recommended to use HuH7 cell line. Therefore, we chose HuH7 and succeeded to generate EB1-KO cell lines.

#8. All data in figure 3 should be shown in comparison to the original HuH7 cells, especially to confirm efficient knockout.

Response: Thank you for your advice. You have raised an important point; however, we generated EB1-KO HuH7 for another reason. Before generating EB1-KO HuH7, we tried to generate an EB1 overexpressing cell line with lentivirus vector. However, our attempt was unsuccessful. So, we changed our mind, and thought generating an EB1 re-expressing EB1-KO cell line would be a substitute for an EB1 overexpressing cell line. Therefore, we compared EB1-KO HuH7 cell lines with EB1 re-expressing EB1-KO HuH7 cell lines without the original HuH7 cell line.

#9. Line 300: Unclear why SLCO1B3 is discussed and for instance NPTX2 and HMSD not. Additionally, the data from the microarray should be validated by at least qPCR before these can be considered reliable enough to thoroughly discuss in line 341 to 359.

Response: Thank you for your comment. NPTX2 (Xu C, et al. NPTX2 promotes colorectal cancer growth and liver metastasis by the activation of the canonical Wnt/B-catenin pathway via FZD6. Cell Death Dis. 2019) and HMSD (Chen F, et al. RNA-seq analysis identified hormone-related genes associated with prognosis of triple negative breast cancer. J Biomed Res. 2020) are also reported to be associated with cancer. We discussed about Dlk1, HAMP, and SLCO1B3, because those genes were the top three of the candidate genes which might be correlated with EB1 expression. We revised the sentence (page 15, line 306 and 307) not to mislead the readers of the journal. Additionally, to validate the microarray results, we performed quantitative real-time PCR. Indeed, the Dlk1 mRNA levels were 6.5-fold higher in EB1-KO HuH7 cells re-expressing EB1 compared with control cells. The data is uploaded as a supplementary figure S4.

#10. Line 363-364 is unclear, this statement requires further explanation.

Response. Thank you for your comment. We have revised the sentences to clarify the first limitation of our study as described below (Page 17, line 370 to 375).

#11. Overall, the Western blot images quality is quite poor and higher resolution images are needed.

Response: Thank you for your advice. We changed some Western blot images (Fig2 and Fig3). It might be better.

---

## [Decision Letter · Decision Letter 1]

26 Aug 2020

PONE-D-20-12629R1

Adenomatous polyposis coli-binding protein end-binding 1 promotes hepatocellular carcinoma growth and metastasis

PLOS ONE

Dear Dr. Yokoo,

Thank you for submitting your manuscript to PLOS ONE. After careful consideration, we feel that it has merit but does not fully meet PLOS ONE’s publication criteria as it currently stands. Therefore, we invite you to submit a revised version of the manuscript that addresses the points raised during the review process.

The rpevious reviewers found that the revised manuscript wa simproved but certain points should be explained further for the publucation.

We look forward to receiving your revised manuscript.

Kind regards,

Jung Weon Lee, Ph.D.

Academic Editor

PLOS ONE

Reviewers' comments:

Reviewer's Responses to Questions

**Comments to the Author**

1. If the authors have adequately addressed your comments raised in a previous round of review and you feel that this manuscript is now acceptable for publication, you may indicate that here to bypass the “Comments to the Author” section, enter your conflict of interest statement in the “Confidential to Editor” section, and submit your "Accept" recommendation.

Reviewer #1: (No Response)

Reviewer #2: (No Response)

2. Is the manuscript technically sound, and do the data support the conclusions?

Reviewer #1: Partly

Reviewer #2: Yes

3. Has the statistical analysis been performed appropriately and rigorously? 

Reviewer #1: Yes

Reviewer #2: Yes

4. Have the authors made all data underlying the findings in their manuscript fully available?

Reviewer #1: Yes

Reviewer #2: Yes

5. Is the manuscript presented in an intelligible fashion and written in standard English?

Reviewer #1: Yes

Reviewer #2: Yes

6. Review Comments to the Author

Reviewer #1: The authors have addressed most of my comments. One key remaining concern is that for most of the statistical analyses, the sample sizes are small due to limited number of replicates (n=3). Although the p-values are often significant, the power of the tests could be small. This may be addressed by stating the limitations of the small sample sizes used in statistical tests, as part of the discussion section.

Reviewer #2: All of my comments have been adequately addressed. However, it would be appreciated if the responses to my comments #1 (explanation of how immunostainings were scored and the cut-off value obtained) and #5 (the concentration of Matrigel used) could also be added to the method section of the manuscript.

7. PLOS authors have the option to publish the peer review history of their article (what does this mean?). If published, this will include your full peer review and any attached files.

Reviewer #1: No

Reviewer #2: No

---

## [Author Response · Author response to Decision Letter 1]

8 Sep 2020

September 8 2020

Jung Weon Lee, Ph.D.

Academic Editor

PLOS ONE

Dear Dr. Lee,

Thank you for your careful consideration of our manuscript (PONE-D-20-12629R1) titled “Adenomatous polyposis coli-binding protein end-binding 1 promotes hepatocellular carcinoma growth and metastasis.” We greatly appreciate the opportunity to submit a revised version for consideration for publication as a Research Article in PLOS ONE. We thank the reviewers whose incisive comments helped us to significantly improve the manuscript. Our point-by-point responses to their comments are stated on the next page. We hope that our paper is now suitable for publication in PLOS ONE.

Sincerely,

Hideki Yokoo

Department of Gastroenterological Surgery I

Hokkaido University Graduate School of Medicine

N-14 W-5, Kita-ku, Sapporo 060-8648, Japan.

E-mail: hi-yokoo@mua.biglobe.ne.jp

Tel: +81-11-706-5927

Fax: +81-11-717-7515.

 

Reviewers' Comments to Author:

Reviewer: 1

Comments and our Responses:

The authors have addressed most of my comments. One key remaining concern is that for most of the statistical analyses, the sample sizes are small due to limited number of replicates (n=3). Although the p-values are often significant, the power of the tests could be small. This may be addressed by stating the limitations of the small sample sizes used in statistical tests, as part of the discussion section.

Response: We revised the Discussion to address the limitations of the small sample sizes as follows: “Although the p values of the results of our experiments indicate significant differences, statistical power may be insufficient because of the limited number of samples. Nevertheless, our data are consistent with findings of others that elevated EB1 expression significantly correlates with the proliferation, migration, and invasion of cancer cells [9, 10, 29].” (Page 18, line 378 to 382)

Reviewer: 2

Comments and our Responses:

All of my comments have been adequately addressed. However, it would be appreciated if the responses to my comments #1 (explanation of how immunostainings were scored and the cut-off value obtained) and #5 (the concentration of Matrigel used) could also be added to the method section of the manuscript.

Response: We added the missing information to the Methods section as follows:

1. “Two independent observers measured the percentage of EB1-positive tumor cells in a blinded fashion and categorized the results into groups ranging (increments of 10%) from 0% to 100%. Based on the average percentage EB1-positivity, if >30% of the HCC cells were stained, we defined the tumor as EB1-positive HCC.” (Page 4, line 81 to 85).

2. “Migration and invasion assays were performed by placing cells into the upper chambers of a Transwell plate (BD Biosciences) without or with 100 µg/cm2 Matrigel coating.” (Page 7, line 137 to 139)

---

## [Editor Report · Decision Letter 2]

8 Sep 2020

Adenomatous polyposis coli-binding protein end-binding 1 promotes hepatocellular carcinoma growth and metastasis

PONE-D-20-12629R2

Dear Dr. Yokoo,

We’re pleased to inform you that your manuscript has been judged scientifically suitable for publication and will be formally accepted for publication once it meets all outstanding technical requirements.

Kind regards,

Jung Weon Lee, Ph.D.

Academic Editor

PLOS ONE
---

## [Editor Report · Acceptance letter]

10 Sep 2020

PONE-D-20-12629R2 

Adenomatous polyposis coli-binding protein end-binding 1 promotes hepatocellular carcinoma growth and metastasis 

Dear Dr. Yokoo:

I'm pleased to inform you that your manuscript has been deemed suitable for publication in PLOS ONE. Congratulations! Your manuscript is now with our production department. 

Kind regards, 

on behalf of

Dr. Jung Weon Lee 

Academic Editor

PLOS ONE